

# Biomass, abundances, and abundance and geographical range size relationship of birds along a rainforest elevational gradient in Papua New Guinea

Katerina Sam[1,2] and Bonny Koane[3]

[1] Biology Centre of the Czech Academy of Sciences, Entomology Institute, Ceske Budejovice, Czech Republic
[2] University of South Bohemia, Faculty of Science, Ceske Budejovice, Czech Republic
[3] The New Guinea Binatang Research Centre, Madang, Papua New Guinea

## ABSTRACT

The usually positive inter-specific relationship between geographical range size and the abundance of local bird populations comes with exceptions. On continents, the majority of these exceptions have been described from tropical montane areas in Africa, where geographically-restricted bird species are unusually abundant. We asked how the local abundances of passerine and non-passerine bird species along an elevational gradient on Mt. Wilhelm, Papua New Guinea relate to their geographical range size. We collected data on bird assemblages at eight elevations (200–3,700 m, at 500 m elevational increments). We used a standardized point-counts at 16 points at each elevational study site. We partitioned the birds into feeding guilds, and we obtained data on geographical range sizes from the Bird-Life International data zone. We observed a positive relationship between abundance and geographical range size in the lowlands. This trend changed to a negative one towards higher elevations. The total abundances of the assemblage showed a hump-shaped pattern along the elevational gradient, with passerine birds, namely passerine insectivores, driving the observed pattern. In contrast to abundances, the mean biomass of the bird assemblages decreased with increasing elevation. Our results show that montane bird species maintain dense populations which compensate for the decreased available area near the top of the mountain.

## INTRODUCTION

Many previous studies have found a positive inter-specific relationship between geographical range size and the abundance of local populations (*Brown, 1984*; *Gaston & Blackburn, 2000*; *Gaston et al., 2000*). The authors hypothesized that (1) species utilizing a wider range or more abundant resources become more abundant and widely distributed (*Brown, 1984*), that (2) high population growth leads to higher abundances and to more occupied sites or that (3) intensive dispersal produces a positive inter-specific abundance-range size relationship (*Borregaard & Rahbek, 2010*; *Gaston et al., 2000*). While there is extensive literature devoted to the patterns of species diversity along elevational gradients

Corresponding author
Katerina Sam, katerina.sam.cz@gmail.com, katerina.sam@entu.cas.cz

(*McCain, 2009*; *Rahbek, 1995*), these studies rarely combine species richness with the study of bird abundance and biomass, arguably more important parameters when it comes to the impact of birds on other trophic levels (but see e.g., *Romdal, 2001*; *Terborgh, 1977*). Even fewer studies have combined these attributes of bird communities with an estimate of available resources (*Ding et al., 2005*; *Ghosh-Harihar, 2013*; *Price et al., 2014*) and/or available area along mountain ranges (e.g., *Ferenc et al., 2016*; *Price et al., 2014*).

Furthermore, many studies do not address the potential differences between passerines and non-passerines or, they completely exclude non-passerine species. *Klopfer & MacArthur (1960)* suggested that phylogenetically younger passerines should be relatively more abundant than non-passerines in unstable environments. They assumed that younger passerines have a less limited central nervous capacity than non-passerines, making them more capable of fitting changing environmental stimuli. In this study, we aimed to test the analogous hypothesis that non-passerines will be more abundant in favorable tropical lowlands with stable climatic conditions than in the more variable environments at higher elevations. In the Himalayas, the ratio of passerines to non-passerines increased very slowly between 160 and 2,600 m a.s.l., and abruptly between ca. 3,000–4,000 m a.s.l. (*Price et al., 2014*) (but note that not all non-passerines were surveyed). Similarly, passerine abundance increased relative to non-passerines with increasing elevation in the Andes (*Terborgh, 1977*). Finally, bird studies focusing on the patterns of abundance or biomass in different feeding guilds along elevational gradients are rare; however, they are essential for improving our understanding of ecosystem dynamics and function.

Macroecological studies have often revealed positive interspecific correlations between geographical range sizes and the abundance of local populations (*Brown, 1984*; *Gaston & Blackburn, 2000*; *Gaston et al., 2000*). The negative correlation between abundance and range-size was showed on temperate datasets (but see *Blackburn, Cassey & Gaston, 2006*) and in montane Africa, where the geographically restricted species are generally more abundant than species with large geographical ranges (*Fjeldså, Bowie & Rahbek, 2012*; *Ferenc et al., 2016*; *Reif et al., 2006*). Several other recent studies of tropical montane taxa report that abundance is uncorrelated with (or negatively correlated to) geographical range size (*Nana et al., 2014*; *Reeve, Borregaard & Fjeldså, 2016*) but see *Theuerkauf et al. (2017)*. The only existing study on this topic from Papua New Guinea showed that abundance was not related to range size (*Freeman, 2018*). However, in contrast to other studies, this one was based solely on mist-netting data and range size was calculated as elevational breath instead of area (*Freeman, 2018*).

The drivers behind the high abundances of montane forest species are unknown. However, several mutually non-exclusive hypotheses have been considered (*Ferenc et al., 2016*). These are: (1) Long-term climatic stability allows specialization of new ecological forms, which then leads to high local abundances of species at mountain tops (*Fjeldså, Bowie & Rahbek, 2012*). (2) Species-poor communities compensate with increased density at high altitudes which then leads to high abundances of montane bird species (*MacArthur, 1972*). (3) Locally abundant tropical montane species have a higher chance of surviving despite their small range sizes while insufficiently abundant species go extinct (*Johnson, 1998*).

To investigate the relationship between abundance and area in different regions, we focused on bird assemblages along the elevational gradient of Mt. Wilhelm in Papua New Guinea. Our goals were to investigate (1) the trends in abundances of birds along the elevational gradient, (2) the changes in relative abundances of different groups of birds (passerines and non-passerines, various feeding guilds), and (3) the effects of geographical range sizes on the abundance of individual species.

## MATERIALS & METHODS

The study was performed along Mt Wilhelm (4,509 m a.s.l.) in the Central Range of Papua New Guinea (Figs. 1A, 1B). The complete rainforest gradient spanned from the lowland floodplains of the Ramu river (200 m a.s.l., 5°44′S 145°20′E) to the treeline (3,700 m a.s.l., 5°47′S 145°03′E; Fig. 1). We completed the study along a 30 km long transect, where eight sites were evenly spaced at 500 m elevational increments. Because of the steep terrain, elevation could deviate by 50 m within each study site. Survey tracks and study sites at each elevation were directed through representative and diverse microhabitats (e.g., ridges, valleys, rivulets; ≥ 250 m from forest edge). In the lowlands, average annual precipitation is 3,288 mm, rising to 4,400 mm at 3,700 m a.s.l. There is a distinct condensation zone at around 2,500–2,700 m a.s.l. (McAlpine, Keig & Falls, 1983). Mean annual temperature typically decreases at a constant rate of 0.54 °C per 100 elevational meters; from 27.4 °C at the lowland site (200 m a.s.l.) to 8.37 °C at the tree line (3,700 m a.s.l.). The habitats of the elevational gradient is described as lowland alluvial forest (200 m a.s.l.), foothill forest (700 and 1,200 m a.s.l.), lower montane forest (1,700–2,700 m a.s.l.), and upper montane forest (3,200 and 3,700 m a.s.l.); according to Paijmans (1976). Plant species composition of the forest (Paijmans, 1976), general climatic conditions (McAlpine, Keig & Falls, 1983) and habitats at individual study sites (Sam & Koane, 2014) are described elsewhere.

Data on bird communities were collected in 2010, 2011 and 2012 during the wet and dry seasons, using a standardized point-counts at 16 points per elevation (Sam & Koane, 2014; Sam et al., 2019). Both visual observations and identifications based on calls were used. The surveys were conducted in the mornings between 0545 and 1100. Each of the 16 sample points had a radius of 50 m (area 0.785 ha per point, which makes 12.56 ha per elevational study site). Points were located 150 m apart to lower the risk of multiple encounters of the same individuals. We visited each point 14 times (8 times during the dry season and 6 times during the wet season). The order of the points was altered during each re-survey, to avoid biases due to time of day. Birds were detected for 15 min during each visit at each point. This resulted in 240 min of daily surveys. During the point-counts, we used a distance sampling protocol. The birds were recorded in five 10-m-wide radial distance bands (Buckland et al., 2001). Detection adjustments, however, proved to come with significant problems in the tropics (Banks-Leite et al., 2014). Therefore, we used the observed abundance only estimates instead of the distance sampling-based estimates in the analyses (see similar reasons and discussion by Ferenc et al., 2016). To evaluate the consistency in our data, we (1) compared the abundances of birds observed during point-counts (reported here) and from mist-netting conducted at the same sites during the same surveys

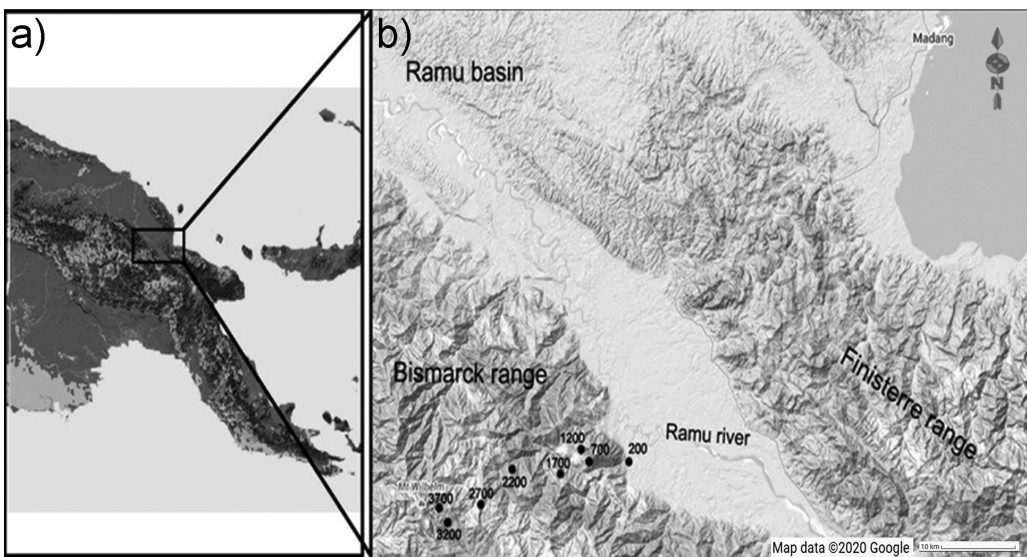

**Figure 1** Location of the elevational gradient of Mt. Wilhelm in Papua New Guinea (A) and the study sites along the gradient (B). Map credit: ©2020 Google.

(*Sam et al., 2019*), (2) we ran all the analyses reported here also with mist-netting data, and we (3) we compared the abundances of the birds recorded during point-counts done in wet and dry season (Figs. S1–S3). The data showed that abundances obtained by mist-netting and by point-counts and by point-counts in wet and dry season are well correlated, and that the trends remain unchanged, when only mist-netting data are used (Figs. S1–S3).

We recorded the number of individuals of each species at any of the 15-min intervals and summed them across all 16 points of each survey day at the certain elevation. Then we averaged these daily abundances across the 14 days (or 6 days of wet season and 8 days of dry season respectively at each elevation). Hereafter we call this measure "*mean elevational abundance*" of a given species at a certain elevational site. After that, we averaged the *mean elevational abundance* across the elevations where the bird species was present to calculate "*mean abundance*" of a given species along the elevational gradient. Species at their elevational range limits usually have low abundances which might be difficult to detect correctly. Therefore, potential errors, which would lead to an erroneous *mean abundance*, might occur if rare individuals at the sites close to their range limits are missed during census. To ensure that our observations are valid, we also repeated the analyses with the "*maximal mean elevational abundance*" of each species (Fig. S4).

To summarize the abundances of bird assemblages at a given elevation (hereafter "*total abundance*") we calculated the sum of the *mean elevational abundance* of all species present at each site (i.e., at 16 points within a 4-hour long survey). Elevations between minimal and maximal range where birds were missing were not considered, i.e., data were not extrapolated, and the birds were given zero abundance at this elevation. The taxonomy used followed the International Ornithological Congress World Bird List version 6.1.

The elevational *weighted mean point* was calculated as the elevation, where the species had potentially the highest abundances. *Weighted mean point* was calculated as a sum of elevations weighed by an abundance of the given species at this elevation which was divided by the sum of the abundances (e.g., Elevation 1 * abundance at elevation 1 + elevation 2 * abundance at elevation 2)/(abundance at elevation 1 + abundance at elevation 2) (Fig. S4). Based on the weighted mean point, we divided the species into three groups as follows: (a) "lowland" group—species with their elevational weighted mean point in the lower part of the elevational gradient (up to 800 m a.s.l.), (b) "middle" group—species with a weighted mean point between 800 and 1,600 m a.s.l., and (c) "montane" group—species with their weighted mean point in the upper third of the gradient (above 1,600 m a.s.l.). Note that a single species (Great cuckoo-dove—*Reinwardtoena reinwardti*), occurring at all sites along the complete gradient between 200 and 3,200 m, thus falls into the group of montane species. To confirm the validity of our data, we also repeated our analyses with the maximal mean elevational abundance point, i.e., the elevational site where we recorded the *maximal mean elevational abundance* (Fig. S4).

All recorded bird species were classified into five trophic guilds: insectivores, frugivores, frugivores-insectivores, insectivores-nectarivores and nectarivores based on dietary information from standard references (*Hoyo et al., 1992–2011*; *Pratt & Beehler, 2015*) and our data (*Sam et al., 2019*; *Sam et al., 2017*). Abundances of passerines and non-passerines and individual feeding guilds were compared by non-parametric Kruskal-Wallis tests. We report the mean ± SE and abundances per 12.56 ha recorded in a 15-minute-long census unless we state otherwise. Geographical range sizes of all birds were obtained from Bird-Life International data zone web pages accessed in July 2016. Bodyweight (mean for males) of the birds were obtained from *Hoyo et al. (1992–2011)*. Bird metabolism was calculated from bodyweight according to available equations (*McNab, 2009*).

We conducted the field work under the Institutional Animal Care and Use Committee approval permit No. 118 000 561 19 and 999 020 778 29 awarded by the PNG National Research Institute permit. Research was also permitted by Australian Bird and Bat Banding permit No. 3173. The data were collected at the land of several rainforest dwelling communities. The customary landowners (Peter Sai, Family Mundo, Alois Koane, Simon Yamah, Samson Yamah, Joe Black) gave fully informed verbal and prior consent to the study of bird communities on their land along Mt. Wilhelm gradient. The negotiations with landowners were organized via The New Guinea Binatang Research Centre, an NGO in Papua New Guinea.

## RESULTS

In total, we recorded 25,715 birds belonging to 249 (Table S1) species from the point-counts along the elevational gradient of Mt. Wilhelm. This represents 87% of bird species recorded along the gradient so far (*Marki et al., 2016*; *Sam & Koane, 2014*; *Sam et al., 2019*). Total bird species richness seemed to show a plateau at lower elevations (up to 1,700 m a.s.l.) and decreased with increasing elevation afterward (Fig. 2A). In contrast, *total abundance* of birds showed a humped shaped pattern, peaking between 1,700 and 2,700 m a.s.l. with ca. 420-450 individuals of all birds per 16 sampling points (i.e., 12.86 ha) (Fig. 2C).
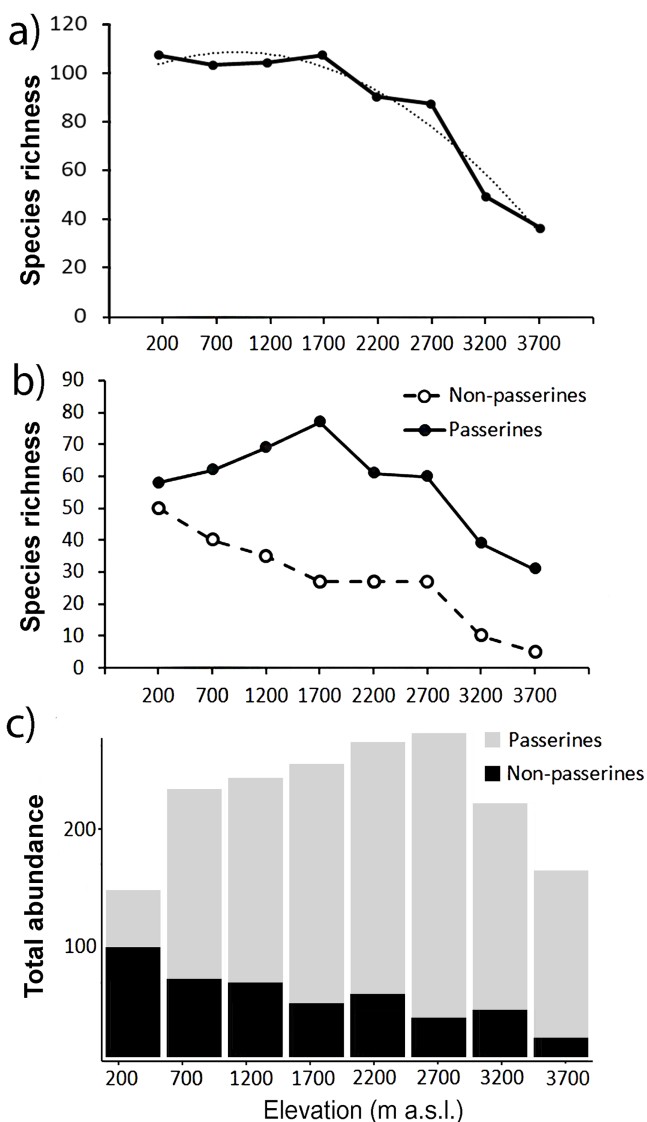

**Figure 2  Patterns of species richness and total abundance of all birds along the elevational gradient of Mt. Wilhel.** Species richness (fitted with exponential function: $y = -2.4107x^2 + 11.756x + 93.946$, $R^2 = 0.95$) of all birds recorded during point-counts from along the elevational gradient of Mt. Wilhelm (A); species richness of passerine and non-passerine birds separately (B). Total (i.e., summed) abundances of passerine (grey) and non-passerine (black) birds at respective elevational sites (C).

## Passerines and non-passerines

Passerines were overall more species rich along the elevational gradient, represented by 161 species in comparison to non-passerines represented by 88 species (Fig. 2B). We observed a linearly decreasing pattern in species richness of non-passerine birds ($N = 8$, $y = -5.9167x + 60.056$, $R^2 = 0.96$) along the elevational gradient and a hump-shaped pattern ($N = 8$, $y = -2.1012x^2 + 18.982x + 27.315$, $R^2 = 0.92$) in species richness of passerine birds (Fig. 2B). The species richness of passerines ($r = 0.52$, $P = 0.19$, $N = 8$) and

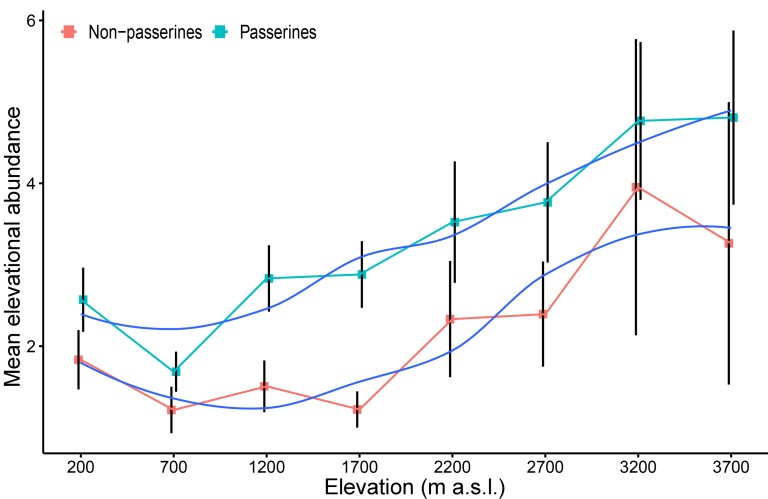

**Figure 3** *Mean elevational abundance* of a passerine and non-passerine bird species (±SE) (i.e., mean number of individuals of a given species at a given elevation) occurring in the particular assemblage along the elevational gradient of Mt Wilhelm (fitted with loess smooth function).

non-passerines ($r = 0.91$, $P = 0.001$, $N = 8$) correlated with their *total abundance* (Figs. 2B and 2C).

The *mean elevational abundance* of passerine birds was overall significantly higher (mean ± SD = 3.90 ± 4.8) than the *mean elevational abundance* of non-passerines (mean ± SD = 2.46 ± 3.1; $W = 21438$; $P < 0.001$). The *mean elevational abundance* of assemblages increased with increasing elevation, with approximately 2.5 times as many individuals per non-passerine species and nearly twice as many individuals per passerine species at the highest elevation than in the lowlands (Fig. 3). The pattern was similar in both wet and dry seasons (Fig. S5). This pattern remains to be valid even when only the *maximal mean elevational abundance* were considered. Birds having their *maximal mean elevational abundance* at higher elevations had abundances higher than birds with maxima in lowlands (Fig. S6)

Passerine birds with an elevational weighted mean point in the montane forest (above 1,600 m a.sl.) had a higher *mean abundance* than those with a middle and lowland distribution (Fig. 4A, Table S1). However, with their increasing elevational weighted mean point, the geographical ranges of the species decreased (Fig. 4B). We found no significant change in the *mean elevational abundance* of non-passerine birds with an elevational weighted mean point (Fig. 4C) but similarly to passerines, non-passerines with a higher elevational weighted mean point had smaller ranges (Fig. 4C). The abundance range-size relationships for all bird species of the complete forestal gradient of Mt. Wilhelm showed a significantly negative relationship ($F_{1,248} = 8.22$, $P = 0.004$, Fig. S7). The trends remained negative, albeit nonsignificant, for passerines ($F_{1,159} = 1.17$, $P = 0.28$) and non-passerines ($F_{1,86} = 2.6$, $P = 0.10$) separately (Fig. S7). However, the relationship of the three bird groups with different elevational weighted mean points showed a variable pattern, as the trend changed from a positive relationship in the lowland group of species, to no trend for
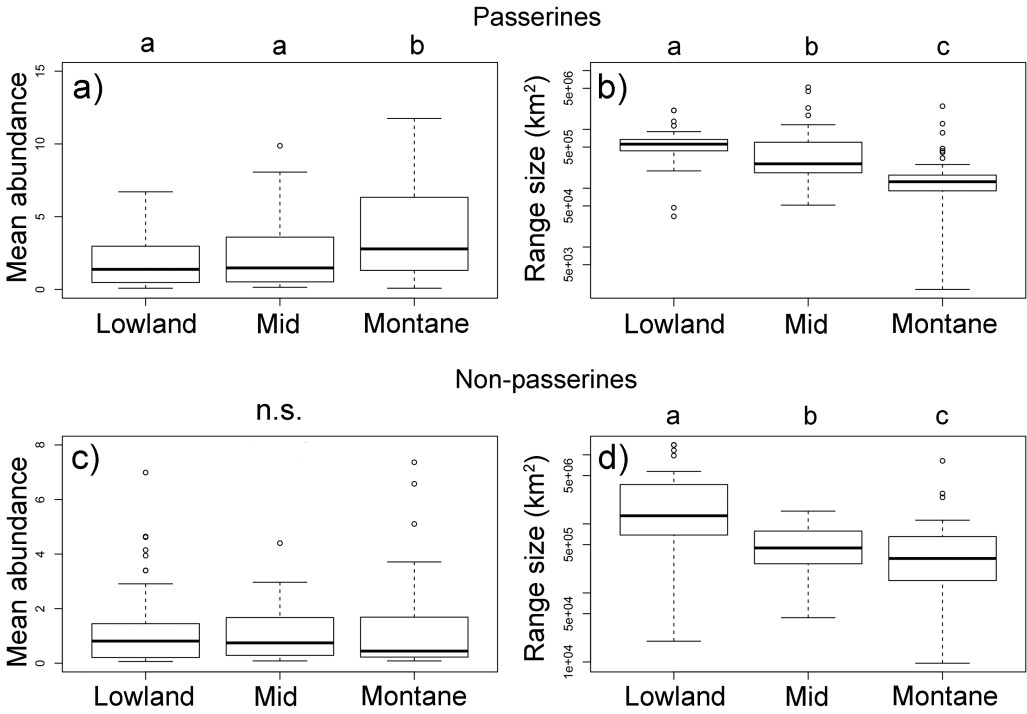

**Figure 4** **Passerine (A, B) and non-passerine (C, F) birds divided into three groups based on the position of their mean-point of elevational distribution on Mt. Wilhelm, and their *mean abundances* (A, C) and geographical range sizes in km$^2$ (B, D).** Kruskal-Wallis—passerines (A) $\chi^2 = 16.3$, $df = 2$, $N = 161$, $P < 0.001$; (B) $\chi^2 = 67.3$, $df = 2$, $N = 161$, $P < 0.001$; non-passerines (C) $\chi^2 = 1.2$, $df = 2$, $N = 88$, $P = 0.549$; (D) $\chi^2 = 19.5$, $df = 2$, $N = 88$, $P < 0.001$. Lowland group = elevational mean-point up to 800 m a.s.l., mid group = elevational mean-point between 801 and 1,600 m a.s.l., and montane group = elevational mean-point above 1,600 m a.s.l.

middle species, and a negative trend for montane species (Fig. S8). The pattern remained similar, when we split the data into abundances in the wet and dry season (Fig. S9). Furthermore, the pattern remains unchanged even when the *maximal mean elevational abundance* is considered in analyses, as the *maximal mean elevational abundance point* and weighted mean point correlated closely (Fig. S10). Finally, more abundant passerine montane birds had not only larger geographical ranges, but also longer elevational ranges (Fig. S11).

## Feeding guilds

Without respect to which feeding guild they belong, species occurring at low elevations usually had a lower *mean elevational abundance* than species occurring at high elevations (Fig. 4A) i.e., their *mean elevational abundance* increased with increasing elevation. Nectarivorous and insectivore-nectarivorous species had the highest *mean elevational abundance* which increased towards higher elevations (Fig. 5A). Within insectivore-nectarivores, the pattern was driven purely by the presence of flocks of nectar-feedings lorikeets at high elevations (i.e., the pattern disappeared when we removed lorikeets from the dataset).

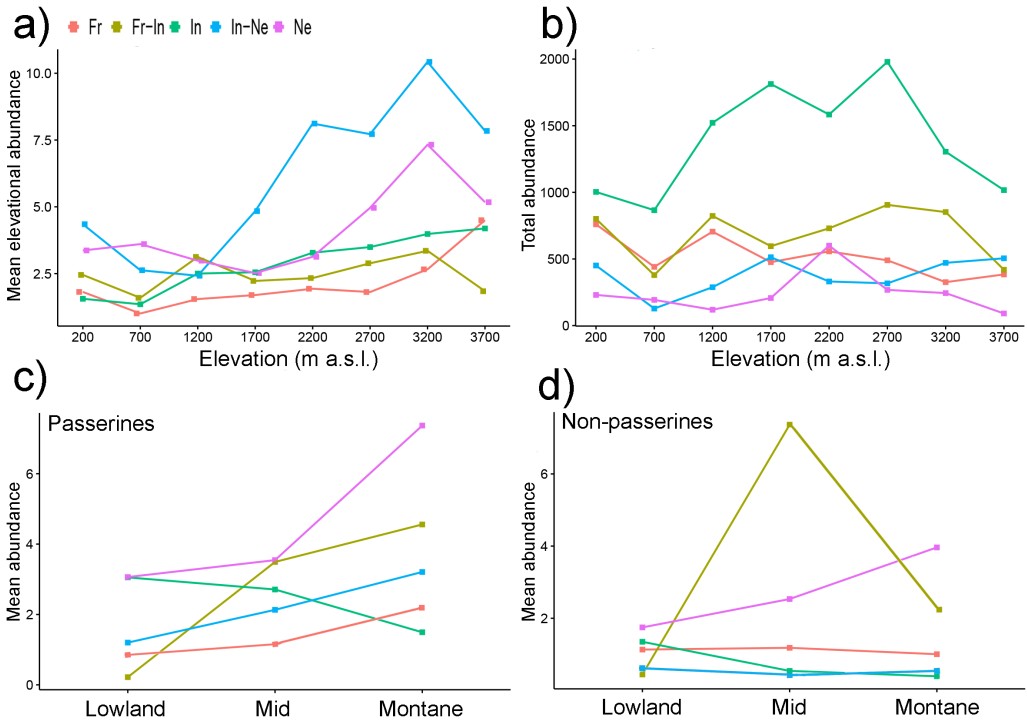

**Figure 5** **Mean elevational abundances of birds partitioned into feeding guilds (A) and** *total abundance* **of bird assemblages partitioned into feeding guilds (B).** *Mean abundances* **of birds partitioned into feeding guilds and into passerines (C) and non-passerines (D).** *Mean elevational abundance* refers to mean number of individuals of a given species at a given elevation. Subsequently, *mean abundance* refers to averaged *mean elevational abundances* of a species across all elevations where it was present. *Total abundance* refers to aggregated abundances of bird assemblage at a given elevations. Ne, Nectarivores; In, Insectivores; In-Ne, Insectivore-nectarivores; Fr, Frugivores; Fr-In, Frugivore-insectivores. Standard errors of the mean are not shown for the clarity of the graph. Lowland group = elevational mid-point up to 800 m a.s.l., mid group = elevational mid-point between 801 and 1,600 m a.s.l., and montane group = elevational mid-point above 1,600 m a.s.l.

*Total abundances* of bird assemblages belonging to different feeding guilds however showed different patterns (Fig. 5B). While *total abundances* of insectivores followed a mid-elevational peak (Fig. 5B), *total abundances* of other feeding guilds showed no trend (Fig. 5B).

Within passerine birds, the *mean elevational abundance* of birds belonging to different feeding guilds (except frugivores) increased with their elevational weighted mean point (Fig. 5C). In contrast, the *mean elevational abundance* of non-passerines birds belonging to various feeding guilds showed various patterns (Fig. 5D).

Mean biomass of bird communities (Fig. 6) recorded at each elevational study site decreased with increasing elevation, thus showing a different pattern from *mean elevational abundance* and *total abundance*. At the two highest elevations (3,200 and 3,700 m) mean biomass of passerines was relatively larger than biomass of non-passerines. The pattern of decreasing biomass was observed both with passerines and non-passerines (Fig. 6A), as well as in all feeding guilds (Fig. 6B). Because large species may have larger ranges

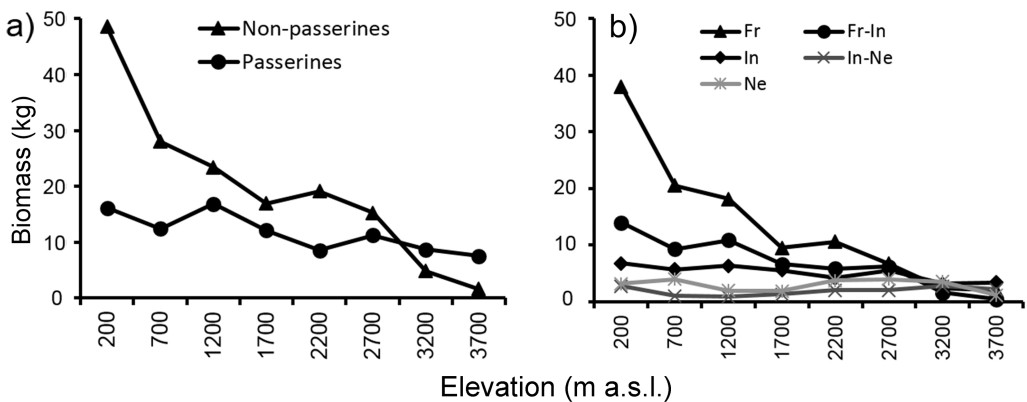

**Figure 6** Mean biomass (across the re-surveys of all point-counts) of passerine and non-passerine birds (A) and birds partitioned into feeding guilds (B) of Mt. Wilhelm (total biomass in kg/12.86 ha).

(*Gaston, 1996a*; *Gaston, 1996b*), we tested how strong the relationship was between body size and geographical range. We found a weakly positive correlation between body size and range size in non-passerines, and no correlation in passerine birds (Fig. S12).

## DISCUSSION

In this study we focused on the patterns and relationships in species richness, abundance and range size in assemblages of passerine and non-passerine birds along a tropical, elevational gradient. After a detectable, initial plateau at low elevations, overall species richness declined with increasing elevation on Mt. Wilhelm (*Sam et al., 2019*), a pattern that is typical for mountains with a humid base (*McCain, 2009*). Further, when considered separately the species richness patterns of passerines (hump-shaped) and non-passerines (steeply decreasing) differed. The findings are, in part, similar to those from Himalaya (*Price et al., 2014*), where species richness of non-passerines decreased with increasing elevation but passerines followed a hump-shaped pattern and their richness peaked at ca. 2,000 m.

*Total abundance* of bird assemblages at individual elevations also follows a different hump-shaped pattern. This mismatch in patterns between observed species richness and abundance is an interesting observation. Previous studies show that unimodal or linearly decreasing patterns in density are usually paralleled by the patterns of total species richness along the same gradients (e.g., *Romdal, 2001*; *Terborgh, 1977*). Our findings are similar to patterns in abundances of birds observed along an elevational gradient in Cameroon (*Ferenc et al., 2016*). In this study, declining species richness was associated with minimal changes in the *total abundance* (i.e., the number of individuals per species) of birds because the average number of individuals per species increased with increasing elevation.

Overall, the trends exhibited by *total abundance* and species richness do not correspond to one another. However, when *total abundance* is partitioned into the hump-shaped trend for passerines and decreasing trend for non-passerine birds, the trends become complementary with their respective species richness. To our knowledge, there is not a

single study focusing separately on the abundance patterns of passerine and non-passerine birds along an elevational gradient. Our data further show that the species richness and abundance of passerines increase relative to non-passerines with increasing elevation. This might be in concordance with previous suggestions that phylogenetically younger passerines should be relatively more abundant in less favorable and unstable environments. *Klopfer & MacArthur (1960)* showed that the proportions of non-passerines to passerines changes from north to south. A study comparable to ours by *Price et al. (2014)* indicated that the ratio between the abundances of passerines and non-passerines increased very slowly between 160 and 2,600 m a.s.l., and then increased abruptly between ca. 3,000–4,000 m a.s.l..

The widespread pattern that abundance is positively correlated with geographic range size (*Gaston & Blackburn, 2000*) does not seem to apply to New Guinean birds distributed along elevational gradients. Contrary to this widely accepted pattern, we described a negative correlation between the local abundance of birds and the complete range size of the given species. The deviation from a positive abundance-area relationship is caused by the combination of the decreasing range size and increasing abundance of birds towards high elevations. This observation is also consistent with the idea of taxon cycles whereby endemic species are confined to mountain tops. This observation also fits the predictions of the density compensation hypothesis in which individual species may increase their abundances to fill the available ecological space in species-poor assemblages (*MacArthur, Diamond & Karr, 1972*). The hypothesis thus assumes that small-range species that have insufficiently sparse local populations become extinct.

We showed that New Guinean bird species with small ranges are associated with high local abundances, as has been suggested for marsupials in Australia (*Johnson, 1998*), birds of the Australian wet tropics (*Williams et al., 2009*) and Afromontane birds (*Ferenc et al., 2016*). There are only a few previous examples of datasets that report either nonsignificant or negative abundance–range-size relationships from birds in temperate zones (*Gaston, 1996a*; *Gaston, 1996b*; *Päivinen et al., 2005*). However, several studies have reported nonsignificant or negative abundance–range-size relationships from the birds in the tropics (*Ferenc et al., 2016*; *Nana et al., 2014*; *Reeve, Borregaard & Fjeldså, 2016*; *Reif et al., 2006*). Although, studies reporting a positive trend (*Theuerkauf et al., 2017*) or no trend (*Freeman, 2018*) in the tropics also exist.

The species richness of birds declined (with a lowland plateau) with increasing elevation on Mt Wilhelm (*Sam et al., 2019*). This is a typical pattern for mountains with a humid base (*McCain, 2009*). However, we found that the number of individuals per bird species increased with increasing elevation and decreasing area. Further investigations of our data and its partitioning into feeding guilds showed that patterns of abundances for passerine birds are driven by insectivorous birds, while frugivores drive the decreasing pattern in non-passerines. This, in turn, is driven solely by the species richness of the feeding guild within the two groups of birds. A high proportion of the non-passerine birds of Mt. Wilhelm are identified as frugivorous (44%) and insectivorous (29%), whereas, most of the passerines (59%) are insectivorous.

The contrasting pattern for the *total abundance* of passerine and non-passerine bird assemblages is an interesting observation considering the decreasing trend in overall environmental productivity (*McCain, 2009*) and food availability (estimated by the abundance of insects and fruits) along the elevational gradient (e.g., *Janzen et al., 1976*; *Loiselle & Blake, 1991*), especially along wet mountains like Mt. Wilhelm (*McCain, 2009*). Along Mt. Wilhelm, abundances of arthropods followed a humped-shaped pattern with a peak at ca. 1,700 m (*Sam et al., 2020*; *Supriya et al., 2019*; *Volf et al., 2020*), providing the best food resources for insectivorous passerines in the middle of the gradient. In contrast, abundance as well as biomass of fruits decreased steeply with increasing elevation (*Hazel, 2019*; *Segar et al., 2017*). Considering that 71% of non-passerines feed on fruits and/or nectar while at least 68% of passerines feed solely on insects, the observed patterns in total abundance might be shaped by the availability of resources. Additionally, abundance patterns in both groups of birds are parallel to the species richness of these groups along our gradient. This corresponds to previously reported results on the relationships between abundance and species richness along elevational gradients (*Terborgh, 1977*).

Mean biomass of bird communities recorded at each elevational study site decreased quite steeply with increasing elevation, showing a different pattern than the *total abundance* of birds at given sites. At the upper most two elevations (3,200 and 3,700 m) mean biomass of passerines was relatively larger than biomass of non-passerines which corresponds, in part, with their *mean elevational abundance* at these elevations. The decrease in bird biomass suggests a decrease in energy flux at given elevations, very likely because of reduction of primary productivity (*Dolton & De Brooke, 1999*).

## CONCLUSIONS

In direct contrast to the abundance-geographical range size relationship hypothesis investigated here, we found that montane species which associated with small geographical ranges have locally higher abundances than lowland species which are associated with large geographical ranges. The *mean abundances* of passerine and non-passerine birds follow a similar trend (significant for passerines, but nonsignificant for non-passerines), with montane birds having higher abundances then lowland birds. Abundances of passerines seem to be driven by insectivores, while non-passerines seem to be driven by frugivores. Our data further show that passerines and non-passerines have different patterns of species richness and *total abundance* along the same elevational gradient.

## ACKNOWLEDGEMENTS

We wish to thank numerous field assistants from Kausi, Numba, Bundi, Bruno Sawmill, Sinopass, and Kegesugl for help in the field and hospitality.

### Funding

The work of Katerina Sam was supported by Czech Science Foundation Grants 18-23794Y and the infrastructure and logistics of the project was financially supported by the European Science Foundation 669609 grant, the Darwin Initiative for the Survival of Species grant 22-002 and the Christensen Foundation grant 2016-8734. The funders had no role in study design, data collection and analysis, decision to publish, or preparation of the manuscript.

### Grant Disclosures

The following grant information was disclosed by the authors:
Czech Science Foundation: 18-23794Y.
European Science Foundation: 669609.
Darwin Initiative for the Survival of Species: 22-002.
Christensen Foundation: 2016-8734.

### Competing Interests

The authors declare there are no competing interests.

### Author Contributions

- Katerina Sam conceived and designed the experiments, performed the experiments, analyzed the data, prepared figures and/or tables, authored or reviewed drafts of the paper, and approved the final draft.
- Bonny Koane performed the experiments, authored or reviewed drafts of the paper, and approved the final draft.

### Animal Ethics

The following information was supplied relating to ethical approvals (i.e., approving body and any reference numbers):

National Research Institut of Papua New Guinea provided Permit No. 11800056119. Australian Bird and Bat Banding provided licence No. 3173.

### Field Study Permissions

The following information was supplied relating to field study approvals (i.e., approving body and any reference numbers):

The data were collected at the land of several rainforest dwelling communities. The customary landowners (Peter Sai, Family Mundo, Alois Koane, Simon Yamah, Samson Yamah, Joe Black) gave fully informed verbal and prior consent to the study of bird communities on their land along Mt. Wilhelm gradient. The negotiations with landowners were organized via The New Guinea Binatang Research Centre, an NGO in Papua New Guinea.

### Data Availability

The raw data is available in the Supplemental File.

## Supplemental Information

Supplemental information for this article can be found online at http://dx.doi.org/10.7717/peerj.9727#supplemental-information.

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
