# Peer review of "Biomass, abundances, and abundance and geographical range size relationship of birds along a rainforest elevational gradient in Papua New Guinea"

_PeerJ, doi:10.7717/peerj.9727_

## Round 0.1 · original submission · Major Revisions

Dear authors, we received two comprehensive reviews. The common denominator is that you need to put more efforts in order to finalize the paper. For example, you need to convince the reader that it is important to use different analysis for avian functional groups, and explain the patterns. Presentation of data is also essential - separating wet and dry seasons data will help the readers. Reviewer 1 has marked the document - please use the suggestions. I hope that after the requested revisions you manuscript will be in much better shape.

·

Basic reporting

see below

Experimental design

see below

Validity of the findings

see below

Additional comments

This paper makes a valuable contribution to our understanding of bird range sizes and diversity along an important tropical elevational gradient. It does, however, need a great deal of work to get it into publishable shape, and I have placed many comments directly on the manuscript. All comments with no entries at all signify typographical errors, of which there are many. Below are some of the major issues, others are left on the MS.

The introduction takes a long time to get to the main question and should start with it.

Two main points:

1. An important issue is what ‘mean elevational abundance’ actually is. The statement is “We recorded the mean number of individuals of a species recorded at 15-min interval (for all 16 points together) as “mean elevational abundance”. How were the 14 visits included? did you sum or divide across the 16 points? In particular why are there no values less than 1.0 in the data? It seems 0s were omitted, but this doesn’t make sense to me. One could also drop both the word mean and elevation from ‘mean elevational abundance’; just say a measure of abundance at a location, which would be clearer.

l.189 the reason total elevational abundance differs from mean elevational abundance took me a very long time to understand, and I am not sure I have got it even now. Please state explicitly what mean elevational abundance is in fig. 4A as everywhere else it is used it refers to individual species. What I think is that fig. 4B is the sum of abundances at an elevation, and fig. 4A is that sum divided by species richness.

2. The second issue is with the estimates of abundance, which were not corrected for detectability. I agree with the authors that this is difficult to do accurately. Nevertheless, I am sure that point estimates will produce some biases in the results (in the E Himalaya for example Ficedula hypereythra is one of the commonest, but most secretive, of birds). I feel that something can be done with your data to at least evaluate how results might be affected. For example, Sam et al. (J Biogeog) present mist-netting results (and perhaps Freeman’s could be used as well). How does relative bird abundance among those species that are readily mist-netted correlate with bird abundance in the point estimates? What happens if you look at range size and mist net estimates? Is there any other way to evaluate consistency of results (e.g. compare wet vs dry seasons when birds may be more or less conspicuous?). As you recorded the species in concentric circles, you could also do some crude detection corrections and see if these make a difference.

The issue of detectability is the reason Price et al. (2014) decided to adopt the British Trust for Ornithology Breeding Bird surveys, and map territories through thorough surveys on 5ha plots. That of course only surveys a fraction of all breeding birds at a location, but gives good estimates of relative abundance, and captures all common species. Note that, despite statements in the paper, all nonpasserines were surveyed and reported in the supplemental data (e.g. Gallus gallus), but they were not analyzed in the paper (indeed as the authors note, the censuses of nonpasserines were not analyzed at all in the paper, we only used inferences from overlapping elevational ranges for that group.) The reason in the overlapping range data several orders were omitted is because they were considered to interact only weakly with other species (more weakly than ants!). Lack in his book ‘Island Biology’ made the same argument.

lines 127-> it is unclear what the difference between the mean point and mid-point is. Is the mean point summed elevations, weighted by abundances?

l. 177 Is it sensible to average abundance across sites, unless one includes all zeros, which would then give an estimate of total abundance on the mountain? Otherwise it seems that the difference between recording a single individual and recording none is very large. I would have thought another reasonable test would be to use simply maximum abundance.

why not test for an association with elevational range as well as geographical range?

why not show a scatter plot of each species mid-point elevation against abundance, and of abundance against geographical range?

Another thing that probably should be done is generating a model of range size vs. abundance + body mass (or if you think abundance is determined by range size, abundance vs range size + body mass. Large species should a priori have larger ranges I think. Although I would certainly not require this for publication, many reviewers would ask you to use phylogenetic corrections in these analyses. Personally, for this paper I don’t think it is needed.

It would be very helpful if the authors were to give a better breakdown in the data supplement: separating wet and dry seasons for example. I think this is quite important to do, and will make the paper much more valuable to other workers going forward.

Trevor Price

Reviewer 2 ·

Basic reporting

The paper describes abundances of birds along a tropical elevation. It employs different perspectives, basically: (i) different classifications of bird species and (ii) combination of geographical range sizes with abundances. Its form fits international scientific standards.

Experimental design

The paper is rather descriptive but it contains tests of relevant hypotheses. It is well elaborated and the data well statistically analyzed. It based on field data, which collection was undoubtedly difficult and demanding work under harsh tropical conditions.

Validity of the findings

The general finding, that montane birds species live in high abundances, was already done elsewhere. However, such evidence is still very limited and replicates are highly appreciated. This paper tries to decompose the question by classification of birds to different groups. It is a possible way to put it forward, even though I do not see too much news after reading the paper. For example, the finding mentioned even in conclusions i.e. that the patterns are driven by insectivores and frugivores just say that these two groups are the most abundant. I guess it would be better to discuss more the exceptions fro the general pattern. I am relatively sceptical about using the overall biomass information as it looks as driven by species richness. However, the conclusions are correct in my opinion, biomass decreases towards high elevations due to relatively lower productivity in the montane regions. The interesting hump-shaped pattern in total abundances is most likely driven by the hump-shaped pattern of species richness. I think this can be discussed.

Additional comments

I like the paper as it provides evidence about high abundances of montane species in the tropics, which is still rare, it provides a thorough analysis makes efforts to put it into the context of available literature. I would suggest putting more efforts into the reasoning why to use different analysis for avian functional groups and later on discuss the differences more, did it help us somehow to understand the pattern?

---

## Round 0.2 · Minor Revisions

Your manuscript has been significantly improved, but it needs additional work to convince the readers that your results are valid. For example, it is important to repeat the analysis using different measures of abundance to demonstrate that your conclusions do not change. It is also important to explain all figures and draw clear conclusion supported by observations. Overall, the paper is promising, and hopefully can be improved to a publishable shape with one more iteration.

·

Basic reporting

see my general comments to the authors. the english still needs work, and there seems to be one major conceptual issue that should be addressed, on the use of mean abundance.

Experimental design

the authors have done a great job at improving the analysis over the first version, but comparing mist net and observational studies and repeating analyses for different seasons.

Validity of the findings

I am sure the findings are valid, but would like to see analyses repeated with max (mean elevational abundance) cf. mean abundance

Additional comments

This is an important paper, presenting much original data and analyses, which will be of great value. I also appreciate the data being presented in a supplement, which still does not happen often enough. The authors have done a nice job and validating their findings through the use of seasonal comparisons and mist net results. The writing could still use some work. I have made some comments on the MS, and on the supplement, both of which I am returning. Major comments, including responses to the rebuttal are below:

I noted one should quote the new Frontiers in Biogeography paper, which the authors could not locate. I meant the one on which Sam is herself a co-author.

One other comment needs addressing. Is it sensible to average abundance across sites, unless one includes all zeros, which would then give an estimate of total abundance on the mountain? Otherwise it seems that the difference between recording a single individual and recording none is very large. I would have thought another reasonable test would be to use simply maximum abundance. – We added the statistical results also for the total abundances, which were actually nearly the same as mean abundances. L 189- 190.

l.133 I still cannot see what mean abundance tells us. It has been used in the main figures (fig4 and 5C,D) as well as several supplemental figures. Supposing 100 birds were recorded in one location and 1 in another, then we would get mean abundance of 50.5. Now suppose that the 1 bird in the second elevation was missed. Then we would get mean abundance of 100. To take a more concrete example, the fact you observed Eclectus roratus occasionally at 1200m results in its mean abundance being reduced from 5.43 to 3.94. In other words, the mere presence of a species across a broader range reduces its mean abundance. At a range limit, species will always go to low abundance.
Eclectus roratus 7.08 3.78 1

It is of interest that total abundance gives a similar result, but I am not sure what the measure is. Is it the summed abundance of an individual species across all elevations? Elsewhere it is defined as summed abundance across all species at one elevation, which makes sense. But if it is summed abundance of an individual species across all elevations, it seems that this also is not the test you want. Supposing you found one species at low abundance across all sites, and one at high abundance at one site. They may have the same total abundance, but surely the test you were looking for was local density vs. range size.

As noted in the earlier review I think you should use maximum (mean elevational abundance) for each species, which is what you use in several places already.

Explain what is mean abundance from mist-netting data (Figure S2). If similar to the census data, then I think it should also be changed.

I am struck with the similarity of these results to those of the east Himalayan gradient, compare fig. 1c of Price et al 2014 with fig. 2b of this paper. The differences are small and perhaps explained by the different census methods. I feel that a reader will be interested in this, rather than a simple dismissal with the statement that the studies are not comparable because not all non-passerines were include. Only 4 non-passerine species, all of low abundance from New Guinea, would not have been included in our censuses (as detailed in a comment in the MS). I guess New Guinea is much less seasonal than 1700m in the east Himalaya, which makes the similarities more impressive.

Trevor Price

---

## Round 0.3 · accepted · Accept

Thank you very much for providing a convincing response, showing solid arguments supporting your methodology, and especially the "mean abundance", that became a focus of discussion.